# Differentiable Sparsification for Deep Neural Networks

## Abstract

Deep neural networks have relieved the feature engineering burden on human experts. However, comparable efforts are instead required to determine an effective architecture. In addition, as the sizes of networks have over-grown, a considerable amount of resources is also invested in reducing the sizes. The sparsification of an over-complete model addresses these problems as it removes redundant parameters or connections. In this study, we propose a fully differentiable sparsification method for deep neural networks, which allows parameters to be zero during training with the stochastic gradient descent. Thus, the proposed method can simultaneously learn the sparsified structure and weights of networks in an end-to-end manner, which can be directly applies to modern deep neural networks and imposes minimum overhead to the training process. To the authors' best knowledge, it is the first fully [sub-]differentiable sparsification method that zeroes out components, and it provides a foundation for future structure learning and model compression methods.

## 1 Introduction

The success of deep neural networks has changed the paradigm of machine learning and pattern recognition from feature engineering to architecture engineering [16, 14, 22, 7, 32]. Although deep neural networks have relieved the burden of feature engineering, comparable human efforts are instead required to determine an effective architecture, such as the number of neurons or layers and the connections between nodes. In addition, as deep neural networks have over-grown (even up to 10–68 million parameters) [8, 10, 15, 32], considerable effort is also being invested in reducing existing model sizes and in meeting the demands of deploying such networks on constrained platforms at inference time [26, 25].

These problems can be addressed by the sparsification of an over-complete model [20]. A network structure can be carved out of an over-complete model by removing redundant blocks [3, 29] or deleting unnecessary connections between nodes or blocks [2, 18, 30], which also reduces the network size. Among several approaches, pruning has long been adapted [17, 6, 26, 19, 5]. It typically requires a pre-trained model and several steps (select unimportant parameters of a pre-trained model, delete the parameters, and retrain the pruned model) and may repeat the process multiple times. Another approach is a sparsity regularizer with the proximal gradient [21, 3, 29, 33] which shrinks redundant parameters to zero during training and requires no pre-trained model. Among the most popular ones is $l_1$-regularizer [28]. However, as it acts on an individual parameter, it often produces unstructured irregular models; thus, it diminishes the benefit of parallel hardware computation, such as GPUs [29]. In order to obtain regular sparse structures, a sparse regularization with $l_2$-norm [3, 29] was adopted on a group of parameters so that all parameters under the same group are either retained or zeroed-out together. By zeroing-out parameters at a group level, the number of neurons or layers can be automatically determined as a part of training. However, the optimization of a regularization term is performed as a separate step separately from the gradient descent-based optimization for prediction

loss. The update rules should be implemented manually and the approach is limited to the cases where closed form solutions for the proximal operation are known.

In this work, we propose a fully [sub-]differentiable sparsification method for deep neural networks, which directly optimizes a regularized objective function and allows parameters to be exactly zero during training with the stochastic gradient descent. Thus, it can simultaneously learn the sparsified structure and weights of deep neural networks in an end-to-end manner. It leads to simpler implementation and it does not require to manually code a pruning step or an update rule like a soft-thresholding operator. It can adopt various norms as a regularizer regardless of whether their closed form solutions for the proximal operator are known or not. Another advantage of the proposed method is that it can be easily applied on a group of parameters or a building block; thus, it can produce a structured model and maximize the benefits of parallel hardware computation (e.g., GPUs) and it well suits the trend of a modularized design in deep learning [22, 27, 7, 32].

# 2   Related Work

## 2.1   Proximal Gradient

Our proposed method is related to sparsity regularization with the proximal gradient [21]. A regularized objective function is written as

$$\mathcal{L}(D, W) + \lambda \mathcal{R}(W), \tag{1}$$

where $\mathcal{L}$ denotes a prediction loss, $\mathcal{R}$ is a regularization term, $D$ is a set of training data, $W$ is a set of model parameters, and $\lambda$ controls the trade-off between prediction loss and model complexity. The most popular regularizer is $l_1$-norm,

$$\mathcal{R}(W) = \sum_i |w_i|, \tag{2}$$

where $w_i$ is an individual element of $W$. To optimize the regularization term, parameter updating is performed with a proximal operator,

$$w_i \leftarrow sign(w_i) \left(|w_i| - \eta\lambda\right)_+, \tag{3}$$

where $\leftarrow$ denotes an assignment operator, $\eta$ is a learning rate, and $(\cdot)_+$ represents $\max(\cdot, 0)$. As this approach acts on an individual parameter, it often produces unstructured irregular models. In order to obtain regular sparse structures, the sparse regularization with $l_{2,1}$-norm can be adopted [3, 29]. All parameters in the same group are either retained or zeroed-out together. The regularization with $l_{2,1}$-norm is written as

$$\mathcal{R}(W) = \sum_g \|\mathbf{w}_g\|_2 = \sum_g \sqrt{\sum_i w_{g,i}^2}, \tag{4}$$

where $W = \{\mathbf{w}_g\}$ and $\mathbf{w}_g$ represents a group of model parameters. The regularization term is optimized with a proximal operator,

$$w_{g,i} \leftarrow \left(\frac{\|\mathbf{w}_g\|_2 - \eta\lambda}{\|\mathbf{w}_g\|_2}\right)_+ w_{g,i}. \tag{5}$$

When a group has only one single parameter, it degenerates to the $l_1$-norm regularization.

Another group regularization is exclusive lasso with $l_{1,2}$-norm [34, 33]. Rather than either retaining or removing an entire group altogether, it promotes the competition or sparsity within a group. The regularization term is written as

$$\mathcal{R}(W) = \frac{1}{2}\sum_g \|\mathbf{w}_g\|_1^2 = \frac{1}{2}\sum_g \left(\sum_i |w_{g,i}|\right)^2, \tag{6}$$

and its updating rule is derived as

$$w_{g,i} \leftarrow sign(w_{g,i}) \left(|w_{g,i}| - \eta\lambda\|\mathbf{w}_g\|_1\right)_+. \tag{7}$$

These proxial operators consist of weight decaying and thresholding steps, and they are performed at every mini-batch or epoch in a sperate step after the optimization of a prediction loss. It requires some extra efforts to implement the update rules manually and applications are limited to the cases where closed form solutions for the proximal operator are known. In contrast, our approach optimizes a regularized objective function directly and allows parameters to be zero during training with the stochastic gradient descent. It leads to simpler implementations and it can employ various norms regardless of whether closed form solutions for proximal operators are known or not, for example $p$-norm with $p < 1$.

## 2.2 Differentiable Approach

Similar to our work, previous differentiable approaches [2, 18] learn the structure of a neural network by optimizing architecture parameters in a relaxed continuous domain, where architecture parameters represent the importance scores of building blocks or the connection strengths between them. However, as the architecture parameter magnitudes cannot be zero during training, the top $k$ connections or components are stochastically or deterministically selected according to the architecture parameter values to derive a discretized architecture. Therefore, the approach may suffer from the discrepancy between a learned architecture and a final discretized one. Moreover, the value of $k$ should be pre-specified manually; thus, the same value is set for all blocks or modules, which may be sub-optimal. Our approach drives the architecture parameters to zero by optimizing a regularized objective function. This can minimize the model discrepancy, and a network can choose the different numbers of components or connections in each module through training.

## 3 Proposed Approach

### 3.1 Base Model

We assume that there are $n$ components in a module. A component can be any building block for a deep neural network or its output. For example, it can be a channel or a layer of convolutional neural networks such as ResNet [7, 8] and DenseNet layer [10]. It can also represent a node in a neural graph [30] or a convolutional neural network [11]. A module represents a composite of components, such as a group of channels or nodes. For illustration purposes, we assume that a module $\mathbf{y}$ can be written as the linear combination of components $\mathbf{f}_i$:

$$\mathbf{y}\left(\mathbf{x}\right) = \sum_{i=1}^{n} a_i \mathbf{f}_i\left(\mathbf{x}; \mathbf{w}_i\right), \tag{8}$$

where $\mathbf{x}$ denotes a module input, $\mathbf{w}_i$ model parameters for component $\mathbf{f}_i$, and $a_i$ an architecture parameter. Model parameters $\mathbf{w}_i$ denote ordinary parameters, such as a filter in a convolutional layer or weight in a fully connected layer. The value of $a_i$ represents the importance of component $i$, and it represents the connection strength between nodes in another context. Enforcing $a_i$ to be zero amounts to removing component $\mathbf{f}_i$ or zeroing-out $\mathbf{w}_i$. Thus, by creating the competition between elements of $a$ and driving them to be zero, we can eliminate unnecessary components or connections. The sample model is simple, but we will show that it can be applied to various cases.

### 3.2 Differentiable Sparse Parameterization

First, we show how to parameterize architecture parameters with non-negative constraints, which is useful for the attention mechanism with the softmax. To set up the competition between the elements of $a$ and allow them to be zero, we parameterize the architecture parameters as follows:

$$\gamma_i = \exp\left(\alpha_i\right) \tag{9}$$

$$\tilde{\gamma}_i = \left(\gamma_i - \sigma\left(\beta\right) \cdot \|\gamma\|_1\right)_+ \tag{10}$$

$$a_i = \frac{\tilde{\gamma}_i}{\sum_{j=1}^{n} \tilde{\gamma}_j}, \tag{11}$$

where $\alpha_i$ and $\beta$ are unconstrained free parameters, $\sigma(\cdot)$ denotes a sigmoid function, and $(\cdot)_+$ represents $relu(\cdot) = \max(\cdot, 0)$. When a parameter is non-negative, the proximal operator of Eq. (7) is reduced to Eq. (10). Although the forms are similar, they have completely different meanings. The proximal operator is a learning rule, whereas Eq. (10) is the parameterized form of architecture parameters, which is part of a neural network.

We can easily verify that $a_i$ is allowed to be zero and is also differentiable from a modern deep learning perspective. The free parameters $\alpha_i$ and $\beta$ are real-valued and they do not restrict a training process with the stochastic gradient descent. Thus, we can train $a_i$ through $\alpha_i$ and $\beta$. The exponential function in Eq. (9) ensures that the architecture parameters are non-negative. Typically, $a_i$ cannot be zero due to the exponential function of Eq. (9). However, $\tilde{\gamma}_i$ in Eq. (10) can be zero by the thresholding operation; hence, $a_i$ can be zero as well. The term $\sigma(\beta) \cdot \|\gamma\|_1$ plays the role of a threshold, and the thresholding operation is interpreted as follows: if the strength of component $i$ in a competition group is small compared to the total strength, it is dropped from the competition. Note that the scalar parameter $\beta$ in Eq. (10), which determines the magnitude of a threshold, is not a hyper-parameter, but its value is automatically determined through training. Mathematically, the thresholding operator is not differentiable, but this should not pose an issue considering the support of $relu$ as a built-in differentiable function in a modern deep learning tool. Additionally, $\gamma$ is non-negative; thus, its $l_1$-norm is simply the sum of $\gamma_i$ (i.e., $\|\gamma\|_1 = \sum \gamma_i$). The softmax of Eq. (11) is also differentiable. The softmax is optional but useful when we need to promote the competition between components as in the attention mechanism.

By relaxing the differentiability, singed architecture parameter can be similarly formed as
$$a_i = sign(\alpha_i)\left(|\alpha_i| - \sigma(\beta)\|\alpha\|_1\right)_+, \tag{12}$$
where $\alpha$ and $\beta$ are free parameters. The gradient of the $sign$ function is zero almost everywhere, but it does not cause a problem for a modern deep learning tool. The equation can be rewritten as

$$a_i = \begin{cases} \left(\alpha_i + \sigma(\beta)\|\alpha\|_1\right)_- & \text{if } \alpha_i < 0 \\ \left(\alpha_i - \sigma(\beta)\|\alpha\|_1\right)_+ & \text{otherwise,} \end{cases}$$

where $(\cdot)_- = \min(\cdot, 0)$. The gradient can be computed separately according to whether its value is negative or not. To the authors' understanding, $sign$ function is already taken care in the explained manner by TensorFlow [1], and thus $tf.math.sign()$ can be simply used without the manual implementation of the conditional statement.

### 3.3 Sparsity Regularizer

In the proposed approach, an objective function is written as
$$\mathcal{L}(D, W, a) + \lambda \mathcal{R}(a), \tag{13}$$
where $a$ denotes the vector of architecture parameters. Sparsifying $a$ is equivalent to sparsifying a deep neural network. Therefore, we can use the regularization term on $a$ to encourage the sparsity of $a$. The proposed method is not limited to a particular norm and we can drive different sparsity patterns depending on the types of norms. For example, the most popular choice for parameter selection is $l_1$-norm, but it is unsuitable on $a$ in Eq. (11) as it is normalized using the softmax. Its $l_1$-norm is always 1, i.e., $\|a\|_1 = \sum_{i=1}^{n} |a_i| = 1$. Therefore, we should employ $p$-norm with $p < 1$:

$$\mathcal{R}(a) = \left(\sum_{i=1}^{n} |a_i|^p\right)^{\frac{1}{p}} = \left(\sum_{i=1}^{n} a_i^p\right)^{\frac{1}{p}}, \tag{14}$$

where the second equality holds as $a_i$ is always non-negative. To the authors' best knowledge, a closed form solution for the proximal operator of $p$-norm with $p < 1$ is unknown, but the regularization term is differentiable almost everywhere as $relu$ is. Thus, the proposed approach can directly optimize the regularized objective function and zero-out components with the stochastic gradient descent.

By simply switching one norm to another one for a regularizer, different sparsity patterns can be derived. For example, an individual component can be removed with $l_1$-norm (Eq. 2) and a group of components or an entire module can be zeroed-out with $l_{2,1}$-norm (Eq. 4). Note that we do not need to manually implement different updating rules as in the proximal gradient approach. We just need to rewrite a regularization term in the objective function. Examples codes and experiment results are shown in the supplementary material.

**3.4 Rectified Gradient Flow**

If $\gamma_i$ Eq. (10) or $\alpha_i$ in Eq. (12) is less than the threshold, the gradient will be zero, and it will
not receive learning signals. However, note that it does not necessarily mean that a component
dies permanently once its importance score is less than the threshold. The component still has a
chance to recover because the threshold is adjustable and the importance scores of other components
may decrease. Nevertheless, to ensure that the architecture parameters of dropped components
continuously receive learning signals, we propose approximating the gradient of the thresholding
function. As in [31] where the gradient of a step function was approximated using that of *leaky relu*
or *soft plus*, we suggest employing *elu* [4] as a variant of the proposed method: *relu* is used in the
forward pass, but *elu* is used in the backward pass.

This heuristic approach leads to a similar learning mechanism proposed in [30], where the gradient
flows to dropped (or zeroed out) edges but does not flow through these dropped edges. The architecture
parameters in our proposed method correspond to the edges in [30]. Note that our approach can
be easily implemented, and it does not require additional codes to control the gradient flow. The
implementation codes are shown in the appendix.

# 4 Application and Experiment

In this section, we show that the proposed approach can be applied to reduce the size of a network as
well as to learn the structure. Our aim is not to achieve state-of-the-art performance but to validate the
idea and the broad applicability of the proposed approach. In order to show the broad applicability
with limited computing resources, we perform experiments with relative small datasets such as
CIFAR-10/100 [13]. Our implementations closely follow those of baseline models, including model
structures and hyper-parameter settings.

## 4.1 Channel Pruning in Convolutional Network

Table 1: Performance on CIFAR-10, DenseNet-100-BC-K12

| Model | Sparsity(%) | | Top-1 Error(%) | | Parmas | | FLOPs | |
|---|---|---|---|---|---|---|---|---|
| | Avg. | Std. | Avg. | Std. | Avg. | Std. | Avg. | Std. |
| Base | 00.0 | 0.0 | 5.44 | 0.11 | $7.6 \times 10^5$ | 0.0 | $5.8 \times 10^8$ | 0.0 |
| NS | 70.0 | 0.0 | 6.53 | 0.19 | $2.9 \times 10^5$ | $9.3 \times 10^2$ | $1.9 \times 10^8$ | $5.0 \times 10^6$ |
| NS | 80.0 | 0.0 | 8.39 | 0.28 | $2.0 \times 10^5$ | $2.0 \times 10^3$ | $1.4 \times 10^8$ | $3.4 \times 10^6$ |
| DS | 70.3 | 0.1 | 5.77 | 0.09 | $2.7 \times 10^5$ | $2.3 \times 10^3$ | $1.8 \times 10^8$ | $1.6 \times 10^6$ |
| DS | 80.4 | 0.1 | 6.64 | 0.11 | $1.7 \times 10^5$ | $0.6 \times 10^3$ | $1.3 \times 10^8$ | $2.0 \times 10^6$ |

Network-slimming(NS) [20] prunes unimportant channels in convolutional layers by leveraging the
scaling factors in batch normalization. Let $x_i$ and $y_i$ be the input and output of batch normalization
for channel $i$ and then the operation can be written as

$$\tilde{x}_i = \frac{x_i - \mu_i}{\sqrt{\sigma_i^2 + \epsilon}}; \ y_i = a_i \tilde{x}_i + b_i,$$

where $\mu_i$ and $\sigma_i$ denotes the mean and standard deviation of input activations, $a$ and $b$ are scale
and shift parameters, $\epsilon$ is a small constant for numerical stability. The scaling parameter $a$ can be
considered as an importance score or an architecture parameter, and the affine transformation can be
re-written as

$$y_i = a \left( \tilde{x}_i + b_i \right).$$

By pushing $a_i$ to be zero, a corresponding channel can be removed. Network-slimming trains an
initial network with $l_1$-regularization on $a$ to identify insignificant channels. After the training,
channels with small values of $a$ are pruned. To compensate the damage caused by pruning, a pruned
network is fine-tuned. In our approach, we parameterize the scaling parameter using Eq. (12) and train
a network with $l_1$-norm on $a$ using the stochastic gradient descent without pruning and fine-tuning.

We perform comparison experiments on CIFAR-10/100 [13]. The training and test sets contain
$50,000$ and $10,000$ samples respectively, and the final test error is reported at the end of training or

fine-tuning on all training images. We adopt a standard data augmentation scheme (random shifting and flipping) as in [7, 8]. In network-slimming, $\lambda$ in Eq. 13 is fixed to $10^{-5}$, but in our approach we vary its value to induce different level of sparsity. DenseNet-BC-K12 with 100 layers [10] and ResNet with 164 layers [7, 8] are employed as base networks. In network-slimming, pre-trained models are obtained by training networks for 160 epochs with the initial learning of 0.1. The learning rate is divided by 10 at 50% and 75% of the total number of training epochs. After the training, channels with small values of $a$ are pruned and a slimmed network is fine-turned for another 160 epochs with the same setting as in the initial training, but learned weights are not re-initialized. In our approach, we train networks for 320 epochs without fine-tuning or re-training. Network-slimming initializes the scaling factor to be 0.5 and we set $\alpha = 0.5 \, (n+1) \, /n$ and $\beta = \log \left( n^2 + n - 1 \right)$ in Eq. 12 so that $a$ starts with 0.5.

Table 1 shows experiment results on CIFAR-10 with DenseNet. More experiments, including ResNet and CIFAR-100, are given in the supplementary materials due to page limitation. The authors strongly urge readers to see the supplementary materials. We ran each experiments 5 times and showed the average and the standard deviation. Our proposed method is denoted by *Differentiable Sparsification* (DS). We controlled the value of $\lambda$ such that similar pruning rate with that of the network-slimming approach. In the tables, sparsity denotes the number of removed channels in hidden layers. The experiments show that the proposed differentiable approach more effectively learns slimed models.

## 4.2 Discovering Neural Wirings

Table 2: Performance on CIFAR-10, Discovering Neural Wirings

| Model | Top-1 Error(%) | | Parmas | | Mult-Adds | |
|---|---|---|---|---|---|---|
| | Avg. | Std. | Avg. | Std. | Avg. | Std. |
| MobileNetV1($\times 0.25$) | 13.44 | 0.24 | $2.2 \times 10^5$ | 0.0 | $3.3 \times 10^6$ | 0.0 |
| No Update($\times 0.225$) | 13.86 | 0.27 | $2.2 \times 10^5$ | $3.7 \times 10^1$ | $4.5 \times 10^6$ | $3.7 \times 10^4$ |
| DNW($\times 0.225$) | 10.30 | 0.20 | $1.8 \times 10^5$ | $6.7 \times 10^1$ | $3.1 \times 10^6$ | $4.6 \times 10^4$ |
| PG-$l_1$-norm | 12.17 | 0.44 | $2.1 \times 10^4$ | $9.4 \times 10^2$ | $3.3 \times 10^6$ | $1.7 \times 10^5$ |
| PG-$l_{1,2}$-norm | 13.62 | 0.56 | $9.6 \times 10^4$ | $1.6 \times 10^4$ | $3.4 \times 10^6$ | $8.6 \times 10^4$ |
| DS-No Rectified Grad. | 10.55 | 0.23 | $6.1 \times 10^4$ | $5.7 \times 10^2$ | $3.4 \times 10^6$ | $4.5 \times 10^4$ |
| DS-Rectified Grad. | 9.36 | 0.27 | $4.7 \times 10^4$ | $8.4 \times 10^2$ | $3.3 \times 10^6$ | $6.7 \times 10^4$ |

Discovering Neural Wirings(DNW) [30] relaxes the notion of layers and treats channels as nodes in a neural graph. By allowing channels to learn connections between them, it jointly discovers the structure and learns the parameters of a neural network. An input to node $v$, $\mathbf{x}_v$, is expressed

$$\mathbf{y}_v = \sum_{(u,v) \in \mathcal{E}} w_{u,v} \mathbf{x}_u,$$

where $\mathbf{x}_u$ denotes the state of a proceeding node, $\mathcal{E}$ represents a edge set and $w_{u,v}$ is a connection weight of an edge. The structure of a neural graph can be determined by choosing a subset of edges.

At each iteration of training, DNW chooses the top $k$ edges with the highest magnitude, which is called a real edge set, and refers to the remaining edges as a hallucinated edge set. On the forwards pass or at inference time, real edges are only used. As DNW allows the magnitude of the weights in both sets to change throughout training, a hallucinated edge may replace a real edge when it strengthens enough. The weights of real edges are updated in an ordinary manner with the stochastic gradient descent, but those of hallucinated edges are updated by a specialized leaning rule: the gradient flows to hallucinated edges but does not flow through them.

The architecture parameters in our proposed method correspond to the edges in DNW. We parameterize the edges using Eq. (12) and train a network with $l_1$-norm on edges to induce sparsity. The rectified gradient leads to an update rule which is similar to that of DNW: the rectified gradient ensures that dropped architecture parameters continuously receive learning signals by approximating the gradient of the thresholding function. However, we do not need to keep track of the real and hallucinated edge sets. We simply optimize the objective function with approximated gradients. The rectified gradient can be implemented in a couple of lines using modern deep learning tools and the code is shown in the supplementary material.

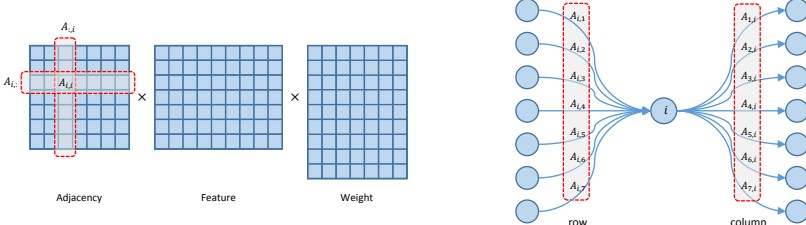

Figure 1: *Left*: Structure of a GCN block. Each block consists of a shared adjacency, an input feature, and a weight matrix. Each row and column of an adjacent matrix are treated as groups to enable the learning of relationship between a node (indexed by $i$) and its neighbors. *Right*: Row grouping creates the competition between in-coming nodes, and column grouping creates the competition between out-going nodes.

We perform experiments on CIFAR-10/100 [13]. The final test error is reported at the end of training without using separate validation data set. MobileNetV1 ($\times 0.25$) [9] is employed as a base model and our implementation closely follows that of DNW. We train for 160 epochs with initial learning rate 0.1. The learning rate is scheduled using Cosine Annealing. DNW chose the value of $k$ such that a final learned model has similar Mult-Adds with the base model, and we also set the value of $\lambda$ in the same manner.

Table 2 shows experiment results on CIFAR-10. We ran each experiments 5 times and showed the average and the standard deviation. Our proposed method is denoted by DS. DNW without the update rule corresponds to DS without the rectified gradient method. Even without the rectified gradient, the performance of the proposed method is close to that of DNW with the update rule. It validates the effectiveness of our approach. We also performed experiments with the proximal gradients of Eq.(3) and Eq.(7), which is denoted by PG in Table 2. PG-$l1$-norm also uses the $l_1$-norm as a regularizer, but the learning is not as effective as ours. Similarly, PG-$l_{1,2}$-norm uses the update rule of Eq.(7) whose shape is similar to our sparse parameterization Eq. (12), but the performance is worse than ours. More experiments, including CIFAR-100, are given in the supplementary materials due to page limitation.

DNW determines the size and the structure of a network by choosing $k$ edges. However, there is no clear notion how to choose $k$ for different stages (or blocks) and thus it uses the same pruning rate for all stages, which may be restrict because each stage may play a different role and need a different amount of resources. In contrast, our approach controls the model complexity by adjusting the value of $\lambda$ in the objective function, and a different amount of resources is allocated for each stage through training. As shown in Table 2, the proposed method uses model parameters more efficiently.

### 4.3 Learning relationship between Nodes in Graph

In this section, we applied the proposed approach to learn the structure of an adjacency matrix in a graph convolutional network (GCN). The purpose of this case study is to test whether our approach can learn semantic structure from data rather than reducing the size of a neural network.

We adopted the model of [11], one of the most successful GCN models. A GCN block or a layer is defined (see Fig. 1) as

$$H^{l+1} = F\left(AH^lW^l\right),$$

where $A$ is an adjacency matrix; $H^l$ and $W^l$ are an input feature and a weight matrix for layer $l$, respectively; and $F$ is a nonlinear activation function. In general, $A$ is non-negative and shared across GCN blocks. It is obtained by normalization. For example, $A = \tilde{D}^{-1}\tilde{A}$ or $A = \tilde{D}^{-\frac{1}{2}}\tilde{A}\tilde{D}^{-\frac{1}{2}}$, where $\tilde{A}$ is an unnormalized adjacency matrix; and $\tilde{D}$ is a diagonal matrix, where $\tilde{D}_i = \sum_j \tilde{A}_{i,j}$. The adjacency matrix represents the connections or relationships between nodes on a graph and is usually given by prior knowledge. Learning the value of $A_{i,j}$ amounts to determining the relationship between nodes $i$ and $j$. If the value of $A_{i,j}$ is zero, it can be considered that the two nodes are unrelated.

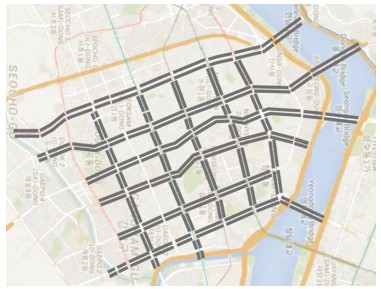

Figure 2: The gray lines represent 170 road links where the experimental data were collected.

As shown in Fig. 1, we defined each row and column as a group. Row grouping creates the competition between in-coming nodes, whereas column grouping creates the competition between out-going nodes. Each row and column of unnormalized adjacency matrix $\tilde{A}$ can be parameterized similarly as in $\tilde{\gamma}$ of in Eq. (10):

$$\gamma_{i,j} = \exp\left(\alpha_{i,j}\right)$$

$$\tilde{A}_{i,j} = \left(\gamma_{i,j} - \sigma\left(\beta_i^r\right) \cdot \left\|\gamma_{i,:}\right\|_1 - \sigma\left(\beta_j^c\right) \cdot \left\|\gamma_{:,j}\right\|_1\right)_+ .$$

The softmax normalization of Eq. (11) is replaced with Sinkhorn normalization [23, 24, 12] to make $A$ is doubly-stochastic: each row and column sum up to 1. Initializing $A$ with $\tilde{A}$, we can convert $\tilde{A}$ into a doubly stochastic matrix by iteratively applying the following equations:

$$A = D_r^{-1} A \quad \text{and} \quad A = A D_c^{-1},$$

where $D_r$ and $D_c$ are diagonal matrices; $[D_r]_i = \sum_j A_{i,j}$; $[D_c]_j = \sum_i A_{i,j}$. Note that although the normalization is iterative, it is differentiable. Balanced normalization is also possible by iteratively applying

$$A = D_r^{-\frac{1}{2}} A D_c^{-\frac{1}{2}} .$$

We verified through numerical experiments that iteratively applying the above equation also makes $\tilde{A}$ to doubly stochastic, but we could not find a theoretical justification. We leave the mathematical proof as an open question for a future work. As competition groups are created in row- and column-wise approaches, a regularized objective function can be written as

$$\mathcal{L}\left(D, W, A\right) + \frac{\lambda}{2} \sum_{i=1}^{N} \left\{ \mathcal{R}\left(A_{i,:}\right) + \mathcal{R}\left(A_{:,i}\right) \right\},$$

where $W = \{W^l\}$, $N$ is the size of square matrix $A$, and $A_{i,:}$ and $A_{:,i}$ denote $i$th row and column vector of $A$, respectively. We employ $l_p$-norm of Eq. (14) with $p = 0.5$ for a regularizer.

To validate our purposed method, we applied a GCN to estimate future traffic speeds in a road network. The traffic speed data were collected from 170 road segments.Thus, the sizes of an adjacent matrix is $170 \times 170$. The map of the data area collected is shown in 2. One-step ahead observation is estimated from eight past observations and an output layer generates 170 estimates, one for each road segment. A prediction loss is measured using the mean relative error (MRE). More detailed specifications of the experimental data and our GCN model can be found in the supplementary material. A prediction loss is measured using the mean relative error (MRE).

Three baseline models were used: two were given the road connectivity and the other was not. For the first baseline, we set the value of $A_{i,j}$ as a constant such that $A_{i,j} = \frac{1}{n_i}$ if node (or road) $i$ and $j$ are adjacent to each other ($n_i$ is the number of neighbors of node $i$), and $A_{i,j} = 0$ otherwise. The second baseline was taken in a similar approach, but we set $\tilde{A}_{i,j} = \exp\left(\alpha_{i,j}\right)$ if node $i$ and $j$ were adjacent to each other to ensure that the strengths of the connections were learned. For the third baseline, the connectivity was not given. However, we set $\tilde{A}_{i,j} = \exp\left(\alpha_{i,j}\right)$ for all $i, j$ regardless of the actual connections. For the proposed method, we parameterized the adjacency matrix as in the

third baseline but applied the sparsification technique. The balanced normalization was applied to all cases except the first baseline, for which the row sum is 1.

To measure the learned relationship between nodes, we propose the following scoring function:

$$\frac{1}{2N} \sum_{i=1}^{N} \sum_{j=1}^{N} \left[ (A^r + A^c) \odot M^k \right]_{i,j},$$

where $A^r = D_r^{-1} A$, $A^c = A D_c^{-1}$, $\odot$ denotes the element-wise product, and $[M^k]_{i,j} = 1$ if the geodesic distance between node $i$ and $j$ is less than or equal to $k$, whereas $[M^k]_{i,j} = 0$ otherwise. The maximum value is 1, and the minimum is 0. For example, the first and second baselines always have the maximum value because their adjacency matrices have exactly the same structure of $M^1$. We calculated the scores for $k = 1$ and 2. Note that we used $A^r$ and $A^c$ instead of $A$ because a sparsified matrix is not guaranteed to be doubly stochastic even if the original Sinkhorn normalization is adopted.

Table 3: Traffic speed prediction with GCN

| Model | #N.Z. | MAPE(%) | L.R.($\times 100$) | | $\lambda$ | #N.Z. | MAPE(%) | L.R.($\times 100$) | |
|---|---|---|---|---|---|---|---|---|---|
| | | | $k = 1$ | $k = 2$ | | | | $k = 1$ | $k = 2$ |
| I | 878 | 5.6623 | 100.00 | 100.00 | 0.050 | 1,220 | 5.4744 | 87.06 | 89.94 |
| II | 878 | 5.5160 | 100.00 | 100.00 | 0.075 | 1,009 | 5.4957 | 88.62 | 91.39 |
| III | 28,900 | 5.6343 | 13.76 | 20.87 | 0.100 | 835 | 5.5336 | 89.79 | 92.13 |
| (a) Baseline models | | | | | (b) Proposed method | | | | |

The performance of the baseline models is shown in Table 4a. We ran each experiment five times and selected the median among the five lowest validation errors. For the first and second baselines, the road connectivity is given, and the number of non-zero elements of the adjacent matrices is 878. Note that the value of the learned relationship for Baselines I and II is constant, but we show it for reference. The road connectivity is not given for the third baseline, and the number of non-zero elements of the adjacent matrix is $28,900 (= 170 \times 170)$. The performance of the proposed model is shown in Table 4b. The experiment of Baseline III shows that a GCN finds a nonlinear mapping between input and target values simply in the way of reducing the prediction loss without learning the semantic relationships between nodes, but the proposed approach finds actual relationships between nodes. We further compared the proposed method with the proximal gradient method. The experimental results are reported in the supplementary material due to page limitation.

## 5   Scope and Limitation

Our aim is not to achieve state-of-the-art performance but to validate the idea and the broad applicability of the proposed approach. To the authors' best knowledge, it is the first fully [sub-]differentiable sparsification method that zeroes out components, and we wish our work would provide a foundation for future structure learning and model compression methods. The limitation of our approach is that a sparsity rate cannot be explicitly specified before training as in conventional pruning approaches. If a specific sparsity rate is required, it should be obtained by a try-and-error. The rectified gradient is effective as shown in the experiments of discovering neural wirings, but it is not clear in which cases it is effective or not. We need more theoretical analysis and leave it for a future work.

## 6   Conclusion

In this study, we proposed a fully differentiable sparsification method that can simultaneously learn the sparsified structure and weights of deep neural networks. Our proposed method is versatile in that it can be seamlessly integrated into different types of neural networks and various problems.

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
