# OpenReview forum: "Differentiable Sparsification for Deep Neural Networks"
_NeurIPS.cc/2021/Conference — NeurIPS 2021 Submitted_

### Official Review · Reviewer_mu5S · 2021-07-16

**Rating:** 4
**Confidence:** 4

**Summary:**

This paper proposes a new differentiable sparsification method to learn sparse structure in neural networks in an end-to-end manner. This objective is straightforward and well-motivated. However, the execution is not novel, comparisons to the SOTA is lacking, reasons for design choices were unsubstantiated with experiments and would thus require additional ablation analysis.


**Limitations And Societal Impact:**

The limitations of the paper is sufficiently discussed in the paper.

**Main Review:**

## Strengths
This paper proposes differentiable sparsification for CNNs and GCNs. The method is straightforward. A potentially novel improvement when compared to DARTS [18] is the inclusion of the proximal operator into the architecture parameter as shown in equation (10) and the evaluation of the backward pass with non-zero surrogate gradients to avoid dead neurons.

## Weaknesses
To begin, Section 2 related work is missing a large chunk of alternative methods that can also sparsity neural networks with end-to-end training. For instance, stochastic networks with policy gradients [b,c], other differentiable approaches to sparsity [d,g], and other more modern baselines for comparison, e.g. for pruning [e,f,h,i].
The majority of Section 2.1 “proximal gradient” bears low relevance to the method proposed in the paper, and should be simplified. How does your method compare to proximal-gradient-based pruning such as [a]?
Section 2.2 has only two references. This subsection require substantial rewrite to include relevant works such as the ones mentioned earlier.

The experiments on CNNs are lacking. The paper has many magical design choices and could necessitate additional ablation and sensitivity analysis. For instance, the choices of $\alpha$ and $\beta$ in Line 204 seem arbitrary. The GCN results are confusing (see Minor issues below), and the reviewer cannot judge its quality in its current presentation.

## Minor issues
* Line 38: “performed as a separate step separately” repetition.
* Line 133: “singed” -> “signed”.
* Line 197: “In network-slimming, λ in Eq. 13 is fixed to $10^{−5}$, but in our approach we vary its value to induce different level of sparsity” It is not clear how this varies the level of sparsity, as in NS the sparsity is mostly determined by $k$ as we keep the top-$k$% of all channels.
* Line 275: grammatical error.
* Table 3: The columns are unexplained. What do you mean by “#N.Z”, “MAPE” and “L.R.”? It is also not clear to the reviewer what models I, II and III mean or how they were defined even after reading the appendix.
* Line 311: Table 4a -> Table 3a.
* Line 317: Table 4b -> Table 3b.

## References:
```
[a]: Li, Yawei et al., Group Sparsity: The Hinge Between Filter Pruning and Decomposition for Network Compression, CVPR 20.
[b] Louizos et al., Learning sparse neural networks through L0 regularization, ICLR 18.
[c] Guo, Shaopeng et al., DMCP: Differentiable Markov Channel Pruning for Neural Networks, CVPR 20.
[d] Kang, Minsoo et al., Operation-Aware Soft Channel Pruning using Differentiable Masks, ICML 20.
[e] Zhuang et al., Neuron-level Structured Pruning using Polarization Regularizer, NeurIPS 20.
[f] Li et al.,  EagleEye: Fast Sub-net Evaluation for Efficient Neural Network Pruning, ECCV 20.
[g] Ning et al., DSA: More Efficient Budgeted Pruning via Differentiable Sparsity Allocation, ECCV 20.
[h] Chin et al., Towards Efficient Model Compression via Learned Global Ranking, CVPR 20.
[i] Guo et al., Multi-Dimensional Pruning: A Unified Framework for Model Compression, CVPR 20.
```


**Time Spent Reviewing:**

5 hours

---

> ### Author Response · Authors · 2021-08-10
> **Clarification and More Experiments**
>
> We appreciate your review comments.
>
> Weakness
>
> 1. Choice of $\alpha$ and $\beta$
>
> First, we are very sorry that there are typos in Line 204. $\alpha$ and $\beta$ should be defined as $\alpha_i=0.5 (n+1)/n$ and $\beta=\sigma^{-1}(\frac{1}{n (n+1)}) = - log (n^2 + n - 1)$, respectively. The equations in Line 204 are derived from Eq. 12 so that we can specify the initial value of $a_i$ through $\alpha_i$ and $\beta$.
> For example, suppose that there are $n=10$ channels in a layer and we want $a_i$ to start with $0.5$, where $a_i$ is a scale parameter in batch-normalization (Line 183 and 184).
> Then, we have $\alpha_i = 0.5 \times 11/10$, $\beta= \sigma^{-1} (\frac{1}{10 \times (10+1)})$ and $\sigma(\beta)=1/(10 \times(10+1))$. If we put the numbers back in Eq 12,
> we have $a_i = \alpha_i - \sigma(\beta) \||\alpha\||_1= \frac{0.5 \times11}{10} - \frac{1}{10 \times 11} \times 10 \times \frac{0.5 \times11}{10} = 0.5 $.
>
> 2. Design choices
>
> Our proposed method is relatively simple to implement and has very small number of additional hyper-parameters.
> For comparison, reference [a] has a complicated fine-tuning process and various additional hyper-parameters.
> For example, reference [a] should decide the following empirically.
>
> a. First, it has to implement an update rule for the matrix $A$.
>
> b. Stop conditions for Algorithm 1 and 2.
>
> c. A hyper-parameter $m$ for a learning rate for matrix $A$ in Section 3.5
>
> d. $\lambda$ for layer balancing in Section 3.6
>
> e. Regularization factor annealing, in Section 3.7
>
> f. Knowledge transfer from a teacher (original) model and hyper-parameters for that, $\alpha$ and $T$ in Section 3.8
>
> 3. Review of previous works.
>
> We really appreciate your pointing to the latest papers. We will update the review section with the references you mentioned.
> To put it simply, we would like to emphasize that most differentiable approaches involve random sampling, but ours does not.
> Our method is simpler to implement and can employ various norms for regularization.  We will more thoroughly review the related works.
> We spared large space for the review of the proximal gradient approach because the proximal gradient method is quite versatile and it is norm-based regularization approach. We expect that our method can substitute for the proximal gradient approach.
>
> 4. Comparison with the latest approaches
>
> We are currently running comparison experiments with [a] and methods referred by [a] as well.
> In the experiment [a], an unpruned baseline model was trained for 300 epochs, and fine-tuned for up-to 500 epochs. Thus, we simply increased 80 epochs (total 400 epochs for end-to-end training) for our approach.
> We borrowed the below comparison from Table 1 in [a].
> For our proposed method, we ran each experiment 5 times and the numbers are the average  from these 5 repetitions.
>
> Model: Accuracy/Baseline (%) - FLOPs (%) - Params (%)
>
> Proposed($\lambda$ = 0.0000275): 5.26/5.13 - 48.29% - 48.41%
>
> Proposed($\lambda$ = 0.0000300): 5.22/5.13 - 47.69% - 46.59%
>
> Hinge [a]: 5.40/4.97 - 53.61% - 70.34%
>
> SSS [b]: 5.78/5.18 - 53.53% - 84.75%
>
> Comparison of CIFAR10/ResNet164 compression results. “FLOPs” and “Params” denote the remaining percentage of FLOP and parameter quantities of the compressed models and the lower the better.
>
>
> Minor Issues
>
> 1. Grammatical error and typos:
>
> We asked a native editor to proof-read our paper after the submission.
>
> 2. Line 197:
>
> Network-slimming (NS) selects top-k channels according to the magnitudes of scaling factors in batch-normalization.
> Our method parameterizes the scaling factors using Eq. (12) and it allows the scaling factors (a_i in Eq. 12) to be zero during training.
> We put regularization on the scaling factors.
> With large value for lambda, an optimization (or training) focuses more on reducing the number of non-zero parameters rather than reducing a prediction loss. For example, please refer to Table 6 and 7 in the PDF supplementary material. As the value of $\lambda$ increases, models get smaller and faster.
>
> 3. Table 3.
>
> We are very sorry that we missed some definitions. We will update the definitions for the terms in a revised paper.
>
>  “#N.Z”: the number of non-zero elements
>
>  “MAPE”: Mean Absolute Percentage Error
>
> "L.R": Learned Relationship, the measure defined between Line 303 and 304.

---

> > ### Comment · Reviewer_mu5S · 2021-08-26
> > **Thanks for the rebuttal, my concerns are not well addressed.**
> >
> > > "For example, reference [a] should decide the following empirically..."
> >
> > I don't think cherry picking [a]
> > for a comparison of hyperparameters
> > is relevant to our discussion.
> >
> > > "... most differentiable approaches involve random sampling, but ours does not."
> >
> > This is simply not true.
> > To begin, the differentiable approaches [d,e,g]
> > mentioned in the review do not involve random sampling
> > for the models.
> > Additionally,
> > even dynamic networks
> > can sparsify themselves with differentiable signals [j,k].
> >
> > > "Table 3."
> >
> > Again, it is still not clear to the reviewer
> > what models I, II and III mean
> > or how they were defined even after reading the appendix.
> > Please address this issue,
> > as the reviewer cannot judge
> > its quality in its current presentation.
> >
> > For the above reasons
> > regarding the novelty and presentation issues,
> > the reviewer maintains the current rating.
> >
> > ```
> > [j]: Hua et al., Channel Gating Neural Networks. NeurIPS 2019.
> > [k]: Gao et al., Dynamic Channel Pruning: Feature Bossting and Suppression. ICLR 2019.
> > ```

---

> > > ### Author Response · Authors · 2021-08-27
> > > **Response to the comments.**
> > >
> > > Thank you for your comments.
> > >
> > > $\>$ I don't think cherry picking [a] for a comparison of hyperparameters is relevant to our discussion.
> > >
> > > Earlier, you said that 'The paper has many magical design choices'. We simply wanted to emphasize that the proposed method is simple and does not have 'magical design choices'.
> > > All pruning/sparsification method have additional hyper-parameters, at least a threshold or $\lambda$ in a regularized objective function. Compare to the previous methods (especially [a]), we have very small number of additional hyper-parameters.
> > > As we explained, the initial values of $\alpha$ and $\beta$ is picked by the simple formula to order to have the scaling factor in batch normalization started with 0.5, which is the same value used in Network-Slimming [19].
> > >
> > > We added code snippets in a pdf appendix and also included source codes in the supplementary material .
> > > If you see any specifically suspicious points, please let us know so that we can explain it more clearly.
> > >
> > > Our only one concern was that whether we should put the ordinary $l_2$-regularizer on $\alpha$ and $\sigma(\beta)$ because we have additional regularization on $a$ in Equation (13).
> > > Initially, we treated $\alpha$ and $\sigma(\beta)$ like any other parameters(i.e, $W$) and put the $l_2$-regularizes on them.
> > > We used the same trade-off weight, i.e, 0.00005, which is equivalent to 0.0001 in PyTorch.
> > >
> > > After the submission and during the discussion period,
> > > we found that putting $l_2$-regularizer on $\beta$ showed a little bit better performance.
> > > For the above additional experiment (ResNet-164/CIFAR100, ResNet-56/CIFAR10),
> > > we put $l2$-regularizer on $\beta$ instead of $\sigma(\beta)$.
> > > In that case of ResNet-164/CIFAR100, trade-off weight 0.00005 was too strong and we set it to 0.00001.
> > >
> > > [19] Zhuang Liu, Jianguo Li, Zhiqiang Shen, Gao Huang, Shoumeng Yan, and Changshui Zhang.
> > > 424 Learning efficient convolutional networks through network slimming. In ICCV, 2017.
> > >
> > > .
> > >
> > > $>$ This is simply not true. To begin, the differentiable approaches [d,e,g] mentioned in the review do not involve random sampling for the models. Additionally, even dynamic networks can sparsify themselves with differentiable signals [j,k].
> > >
> > > To our understanding, [d] and [g] involve random sampling for the models,
> > >
> > > [d] Kang, Minsoo et al., Operation-Aware Soft Channel Pruning using Differentiable Masks, ICML 20:
> > > Please take a look at Equation (7) and related comments in the paper. "where $g_0$ and $g_1$ are samples drawn from Gumbel(0,1) distribution"
> > >
> > > [g] Ning et al., DSA: More Efficient Budgeted Pruning via Differentiable Sparsity Allocation, ECCV 20:
> > > Please take a look at Figure (3) and its captions: 'Then, the channel-wise masks $m_i$ are sampled"
> > >
> > > [e] Zhuang et al., Neuron-level Structured Pruning using Polarization Regularizer, NeurIPS 20:
> > > It is not a differentiable approach. It is pruning method which requires a threshold and a fine-tuning stage.
> > > From Section 3.3 Pruning Strategy, "We still need a threshold value to prune away neurons with small scaling factors." and "After pruning, we fine-tune the pruned network on the training data."
> > >
> > >
> > > .
> > >
> > >
> > > $>$ "Table 3."
> > >
> > >  I am really sorry for your confusion. The models I, II and III are baselines and they are explained in Line 294-302 of the main paper.

---

### Official Review · Reviewer_6Lqs · 2021-07-17

**Rating:** 5
**Confidence:** 4

**Summary:**

This paper presents a network sparsification method. It represents a network as a linear combination of different components, and the weights for different components are parameterized by some unconstrained parameters. By enforcing the Lp norm on the weights of different components, all parameters can be learnt in an end-to-end manner. Then the proposed method is applied to three applications: channel pruning in CNN, Discovering Neural Wirings, and Learning relationship between nodes in Graph. Experiments validate the effectiveness of the proposed solution.

**Ethical Concerns:**

No.

**Limitations And Societal Impact:**

Yes.

**Main Review:**

+The idea of representing the network as a linear combination of different components and sparsify the weights of different components is interesting.
+The idea of parameterize the weights with unconstrained parameters is interesting.
+The paper is readable and easy to follow.


- The proposed method imposes an Lp norm on the weights. However, in the first application (Channel Pruning), p is set to 1. while in the third application, p is set to 0.5. As stated in the section 3.3, the line before equation 14, p should be less than one. Then why you use p=1 in the first application? Then how different p's affect the final performance?
- Regarding the baselines in Channel pruning, there are many state-of-the-art methods, however, only NS is compared and it is a relative out-of-date method. More recent methods should be compared.
- Regarding the first and second applications, only a small dataset is used. More challenging dataset, like ImageNet should be used to further validate the performance of different methods. As reported in previous work, the performance of different methods on different datasets sometimes is not consistent.
- As  lambda in equation 13 is an important factor for evaluating the sparsity of the network. Then please show how different lambda's affect the performance in the first and second applications.

**Time Spent Reviewing:**

3

---

> ### Author Response · Authors · 2021-08-10
> **Clarification and More Experiments**
>
> We are sorry for your confusion, and we would like to give your more clear explanation.
>
> 1. Lp-Norm.
>
> 1.1. For the channel pruning of Section 4.1, we set p to 1 simply because the baseline (Network-Slimming) used the L1-norm. We tried to keep the network structure and hyper-parameter setting the same as the baseline.
>
> 1.2. For the third application, the L1-norm is always one because the adjacent matrix is normalized using the Sinkhorn normalization, which can be considered as matrix-version softmax. Similarly, in Section 3.3 we wanted to point-out that L1-norm does not work for the softmax.
>
> 1.3. We agree with you that  different p's affect the final performance but we have limited computing resources and we could not try other norms.
>
>
> 2. Latest baselines
>
> Another reviewer pointed-out the same thing as well, and we agree with you.
> We are currently performing comparison experiments with [a] and other methods listed in the reference.
> The reference [a] is given by another reviewer.
> The followings are experiment results up to now, and we will report more experiments during the rebuttal period and in the final paper.
> We borrowed the below comparison from Table 1 in [a]. In the experiment [a], an unpruned baseline model was trained for 300 epochs, and fine-tuned for up-to 500 epochs. Thus, we simply increased 80 epochs (total 400 epochs for end-to-end training) for our approach.
> For our proposed method, we ran each experiment 5 times and the numbers are the average  from these 5 repetitions.
>
> Model: Accuracy/Baseline (%) - FLOPs (%) - Params (%)
>
> Proposed($\lambda$ = 0.0000275): 5.26/5.13 - 48.29% - 48.41%
>
> Proposed($\lambda$ = 0.0000300): 5.22/5.13 - 47.69% - 46.59%
>
> Hinge [a]: 5.40/4.97 - 53.61% - 70.34%
>
> SSS [b]: 5.78/5.18 - 53.53% - 84.75%
>
> Comparison of CIFAR10/ResNet164 compression results. “FLOPs” and “Params” denote the remaining percentage of FLOP and parameter quantities of the compressed models and the lower the better.
>
> [a]: Li, Yawei et al., Group Sparsity: The Hinge Between Filter Pruning and Decomposition for Network Compression, CVPR 2020
>
> [b]: Zehao Huang and Naiyan Wang. Data-driven sparse structure selection for deep neural networks. In Proc. ECCV, 2018.
>
> 3. Experiments on large dataset: ImageNet.
>
> We would like to ask your understanding that we have very limited computing resources. After the submission, we performed DWN [30] experiment on ImageNet. Our proposed method achieved similar Top-1 Accuracy and Multi Add, but had less parameters. More comparison experiments will be given during the rebuttal period and in the final paper. We borrowed the below comparison from Table 3 in [30].
>
> [30] Mitchell Wortsman, Ali Farhadi, and Mohammad Rastegari. Discovering neural wirings. In 448 NeurIPS, 2019.
>
> Proposed Method
>
> Lambda: Params - Mult Adds - Top-1 Accuracy
>
> $\lambda$=0.0000125: 1.67M - 161M - 70.46 %
>
> $\lambda$=0.0000150: 1.62M - 154M - 70.55 %
>
> $\lambda$=0.0000175: 1.56M - 147M - 69.80 %
>
> MobileNetV1-DNW(x0.49): 1.8M - 154M - 70.4%
>
> 4. Lambda
>
> The lambda of the second application is given in the PDF supplementary material. The lambda of Table 1 in Section 4.1 are 0.0000551 and 0.0001040. We will update the lambda of the first application further in the final paper.

---

### Official Review · Reviewer_XHsq · 2021-07-17

**Rating:** 5
**Confidence:** 4

**Summary:**

This study proposes a method that enables differentiable sparsification for deep neural network and can be applied into model compression and architecture learning tasks. Experiments on small datasets are conducted to demonstrate the effectiveness of reducing network sizes and learning network architectures.

**Limitations And Societal Impact:**

I have no concern with the limitations and societal impact of the study.

**Main Review:**

The proposed method is simple. The paper has a clear demonstration. However, this study lacks novelty. The core contribution is an engineering practice composed of the three steps in Eq. 9 - Eq. 11, and the regularization term in Eq. 14. There are variant ways to achieve differentiable zero weights. What is and how to verify the superiority of the proposed method in Eq. 9 – Eq. 11? In experiments, only analytical experiments on small datasets are conducted. There is no comparison with other model compression or architecture search methods on challenging tasks. Therefore, the potential applicability of this study is not convincing.

**Time Spent Reviewing:**

5

---

> ### Author Response · Authors · 2021-08-10
> **Additional experiments on ImageNet**
>
> We appreciate your review comments.
>
> 1. The proposed method is simple:
>
> We think that the simplicity is the best advantage of our proposed work. It is easily applicable to various DNN models and problems.
> In a field, a simplicity is often preferred over a performance. As you mentioned, there are various approaches, so we think that practitioners would try the simplest one first.
> In addition, we would like to emphasize that it is the first fully [sub-]differentiable sparsification method even if it is simple.
>
> 2.  Eq. 9 - Eq. 11:
>
> The parameterization has broad applicability as shown in the paper.
> It can be applied to structured sparsification as well as unstructured sparsification. Most of all, it is applicable to various norm-based regularization approaches. In case of the proximal gradient, it requires to derive the soft-thresholding steps analytically, but some norms do not have closed form solutions for the  proximal operator, such as l_p norm with p < 1 of Eq. 14.  Thus, it has limited applicability in that point, but ours does not. Please refer to Section 1 and Section 5.3 of the appendix (PDF supplementary material) for additional empirical studies. We believe that the proposed method can substitute for the proximal-gradient approach.
>
>
> 3. Experiment:
>
> We would like to ask your understanding that we have very limited computing resources.
> After the submission, we performed DWN [30]  experiment on ImageNet.
> Our proposed method achieved similar Top-1 Accuracy and Multi Add, but had less parameters.
> More comparison experiments will be given during the rebuttal period and in the final paper.
> We borrowed the below comparison from Table 3 in [30].
>
> [30] Mitchell Wortsman, Ali Farhadi, and Mohammad Rastegari. Discovering neural wirings. In 448 NeurIPS, 2019.
>
>
> Proposed Method for ImageNet
>
> Lambda: Params - Mult Adds - Top-1 Accuracy
>
> $\lambda$=0.0000125: 1.67M - 161M - 70.46 %
>
> $\lambda$=0.0000150: 1.62M - 154M - 70.55 %
>
> $\lambda$=0.0000175: 1.56M - 147M - 69.80 %
>
>
> Previous Methods
>
> Model: Params - Mult Adds - Top-1 Accuracy
>
> MobileNetV1 (x0.5): 1.3M - 149M - 63.7%
>
> MobileNetV2 (x0.6): NA - 141M - 66.6%
>
> MobileNetV2 (x0.75): NA - 145M - 67.9%
>
> DenseNet (x1): NA  - 142M - 54.8%
>
> Xception (x1): NA - 145M - 65.9%
>
> ShuffleNetV1 (x1, g=3): NA - 140M - 67.4%
>
> ShuffleNetV2 (x1): 2.3M - 146M - 69.4%
>
> MobileNetV1-RG(x0.49): 1.8M - 170M - 64.1%
>
> MobileNetV1-DNW(x0.49): 1.8M - 154M - 70.4%

---

> > ### Comment · Reviewer_XHsq · 2021-08-20
> > **reply to the response**
> >
> > Thanks for the response and experiments. I understand the difficulty of limited computing resource. Performance on challenging tasks is not a decisive factor. But as indicated by multiple reviewers, the current technical contribution is minor. In this case, I am not convinced of the advantage of simplicity of the proposed method when its experimental performance is not solid enough. So I tend to hold the view that this is a borderline paper and keep my original rating.

---

> > > ### Author Response · Authors · 2021-08-22
> > > **Question regarding the comments:**
> > >
> > > Thank you for your comments.
> > >
> > > I am sorry, but could you clarify the meaning of  'when its experimental performance is not solid enough'?
> > >
> > > Do you mean that we should perform
> > >
> > > 1. experiments on large scale data such as ImageNet, or
> > >
> > >
> > > 2. experiments on some other tasks, such as image segmentation and object detection rather than classification, or
> > >
> > >
> > > 3. comparison experiments with previous model compression methods?

---

> > > > ### Author Response · Authors · 2021-08-29
> > > > **Additional Comparison Experiments**
> > > >
> > > > We have performed additional experiments to validate the propose method and
> > > > compared it with previous several methods.
> > > > Please, take a look at the experimental results.
> > > >
> > > > ResNet-164/CIFAR100: https://openreview.net/forum?id=_x4A8IZ-rRv&noteId=cbua9busXR
> > > >
> > > > ResNet-56/CIFAR10: https://openreview.net/forum?id=_x4A8IZ-rRv&noteId=mYp0fwLupVB

---

### Official Review · Reviewer_bkKy · 2021-07-18

**Rating:** 5
**Confidence:** 4

**Summary:**

This paper proposes a component-wise network sparsification framework based on proximal gradient descent. In the proposed method, each component is associated with a learnable coefficient which is further regularized through a regularizer. The coefficients learning, based on proximal gradient methods, will be projected/thresholded to zero, indicating the component is dropped. Authors also adopt some tricks to relax the backward gradient flow to ensure the components are not dead always. The experiments show some analysis of the proposed method on channel pruning, discovering neural wirings, and graph structure learning.

**Limitations And Societal Impact:**

No obvious potential negative societal impact

**Main Review:**

originality:
The proposed method in the paper is a quite straightforward extension of previous works which use proximal gradient descent to perform network sparsification. The main difference is that the target variable to be regularized is changed to some component coefficients.

quality:
The proposed method should be technically sound. However, it seems that there are several potential concerns
- Although the authors mention that normal sparsification would lead to irregular structures, it seems that there are no experiments to validate that irregular structures could potentially cause some problems. Neural networks are normally randomly initialized with each weight potentially swappable. It's hard to see how this dropping component intuition would be very different/useful in network sparsification.
- The component size also needs to be specified in prior manually, which introduces another limitation in the proposed method.

clarity: The paper's writing is pretty clear. A small typo: line 133 "singed"

significance: The proposed method is a simple application of proximal gradient-based sparsification. Doesn't seem like this is a significant method for related fields.

**Time Spent Reviewing:**

1.5 - 2 hours (including reading related works)

---

> ### Author Response · Authors · 2021-08-10
> **Our proposed method is related to the proximal gradient, but does not belong to it.**
>
> We appreciate your review comments.
>
> First, we would like to emphasize that our proposed method is related to the proximal gradient, but not belong to the category. Our method directly optimizes a regularized objective function using the SGD. It does not require to manually implement a learning or an update rule, such as soft-thresholding steps, Eq. 3, 5 and 7 in our paper. In contrast, the proximal-gradient approach requires to derive the soft-thresholding steps analytically and to implement them manually.
>
> 1. Weight Pruning and Proximal Gradient with L1-norm:
>
> It is well known that weight pruning and l1-regularization on an individual parameter results in irregular structures. Please refer to the introduction of [29].  The structured sparsifcation is preferable in that it can maximize the benefits of parallel hardware computation (e.g., GPUs). Our method is applicable to unstructured and structured sparsification. Channel pruning of Section 4.1 belongs to the structured sparsification and DNW of Section 4.2 can be considered as unstructured sparsification.
>
> [29] Wei Wen, Chunpeng Wu, Yandan Wang, Yiran Chen, and Hai Li. Learning structured sparsity in deep neural networks. In NIPS, 2016.
>
> 2. Component Size:
>
> The component size needs not be specified manually. For example, in the channel pruning of Section 4.1, $n$ of Eq (8) is given as the number of channels in each layer, which comes from an original model.
>
> 3. Clarity:
>
> Thank you for your point-out.
> We asked a native editor to proof-read our paper after the submission. We believe the readability was much improved.

---

### Author Response · Authors · 2021-08-18
**Additional comparison experiments on ResNet-56, CIFAR10**

We borrowed the below comparison table from Table 1 in [a].
In the experiment [a], an unpruned baseline(or a teacher) model was trained for 300 epochs, and fine-tuned for maximum 500 epochs. Thus, we simply trained a slim model for 480 epochs. For our proposed method, we ran each experiment 5 times and the numbers are the average from these 5 repetitions.

.

Comparison of CIFAR10/ResNet-56 compression results. “FLOPs” and “Params” denote the remaining percentage of FLOP and parameter quantities of the compressed models and the lower the better.

.

Method$\space\space\space\space\space\space$ | Top-1/BL (%) | FLOPs (%) | Params (%)

[56]$\space\space\space\space\space\space\space\space\space\space\space\space$ |$\space\space\space$ 7.74/6.96 $\space\space$|$\space\space\space\space$ 79.70$\space\space\space$ |$\space\space\space$ 79.51

GAL-0.6[30] |$\space\space\space$ 6.62/7.64 $\space\space$|$\space\space\space\space$ 63.40$\space\space\space$ |$\space\space\space$ 88.20

[25]$\space\space\space\space\space\space\space\space\space\space\space\space$ |$\space\space\space$ 6.94/6.96 $\space\space$|$\space\space\space\space$ 62.40$\space\space\space$ |$\space\space\space$ 86.30

NISP[51] $\space\space\space\space\space$|$\space\space\space$ 6.99/6.96 $\space\space$|$\space\space\space\space$ 56.39$\space\space\space$ |$\space\space\space$ 57.40

CaP[35] $\space\space\space\space\space\space$|$\space\space\space$ 6.78/6.49 $\space\space$|$\space\space\space\space$ 50.20$\space\space\space$ |$\space\space\space$ N/A

ENC[21]$\space\space\space\space\space\space$|$\space\space\space$ 7.00/6.90 $\space\space$|$\space\space\space\space$ 50.00$\space\space\space$ |$\space\space\space$ N/A

AMC[14]$\space\space\space\space\space\space$|$\space\space\space$ 8.10/7.20 $\space\space$|$\space\space\space\space$ 50.00$\space\space\space$ |$\space\space\space$ N/A

KSE[28] $\space\space\space\space\space\space\space$|$\space\space\space$ 6.77/6.97 $\space\space$|$\space\space\space\space$ 48.00$\space\space\space$ |$\space\space\space$ 45.27

FPGM[15]$\space\space\space\space$|$\space\space\space$ 6.74/6.41 $\space\space$|$\space\space\space\space$ 47.70$\space\space\space$ |$\space\space\space$ N/A

Hinge[a] $\space\space\space\space\space$|$\space\space\space$ 6.31/7.05 $\space\space$|$\space\space\space\space$ 50.00$\space\space\space$ |$\space\space\space$ 48.73

.

Our Proposed method.

No Distill.$\space\space\space\space\space$|$\space\space\space$ 6.67/5.69 $\space\space$|$\space\space\space\space$ 49.83$\space\space\space$ |$\space\space\space$ 51.89

With Distill.$\space\space$ |$\space\space\space$ 6.07/5.69 $\space\space$|$\space\space\space\space$ 51.58$\space\space\space$ |$\space\space\space$ 61.76

.

Our Baseline/Teacher Model (ResNet-56)

Method$\space$ | Top-1 (%) $\space\space$ | FLOPs $\space\space\space\space\space\space$  | Params

Baseline | 5.69 $\pm$ 0.14 | $2.52 \times 10^8$ | $8.53 \times 10^5$

.

Detailed Results for Proposed Method - without distillation

$\lambda$=0.0000250$\space$|$\space\space\space$ 6.38/5.69 $\space\space$|$\space\space\space\space$ 57.62$\space\space\space$ |$\space\space\space$ 58.24

$\lambda$=0.0000375$\space$|$\space\space\space$ 6.35/5.69 $\space\space$|$\space\space\space\space$ 55.91$\space\space\space$ |$\space\space\space$ 55.29

$\lambda$=0.0000500$\space$|$\space\space\space$ 6.67/5.69 $\space\space$|$\space\space\space\space$ 49.83$\space\space\space$ |$\space\space\space$ 51.89

$\lambda$=0.0000625$\space$|$\space\space\space$ 6.63/5.69 $\space\space$|$\space\space\space\space$ 47.16$\space\space\space$ |$\space\space\space$ 49.21

$\lambda$=0.0000750$\space$|$\space\space\space$ 6.84/5.69 $\space\space$|$\space\space\space\space$ 45.27$\space\space\space$ |$\space\space\space$ 47.10

.

Detailed Results for Proposed Method - with distillation, $\alpha=0.4$ and $T=4$ in Eq. (9) in [a].


$\lambda$=0.0001625$\space$|$\space\space\space$ 5.90/5.69 $\space\space$|$\space\space\space\space$ 56.58$\space\space\space$ |$\space\space\space$ 66.91

$\lambda$=0.0001750$\space$|$\space\space\space$ 5.97/5.69 $\space\space$|$\space\space\space\space$ 54.79$\space\space\space$ |$\space\space\space$ 64.18

$\lambda$=0.0001875$\space$|$\space\space\space$ 6.07/5.69 $\space\space$|$\space\space\space\space$ 51.58$\space\space\space$ |$\space\space\space$ 61.76

$\lambda$=0.0002000$\space$|$\space\space\space$ 5.99/5.69 $\space\space$|$\space\space\space\space$ 51.56$\space\space\space$ |$\space\space\space$ 61.11

$\lambda$=0.0002125$\space$|$\space\space\space$ 6.08/5.69 $\space\space$|$\space\space\space\space$ 48.59$\space\space\space$ |$\space\space\space$ 59.05

.

[a]: Li, Yawei et al., Group Sparsity: The Hinge Between Filter Pruning and Decomposition for Network Compression, CVPR 2020

[56] Chenglong Zhao, Bingbing Ni, Jian Zhang, Qiwei Zhao, Wenjun Zhang, and Qi Tian. Variational convolutional neural network pruning. In Proc. CVPR, pages 2780–2789, 2019.

[30] Shaohui Lin, Rongrong Ji, Chenqian Yan, Baochang Zhang, Liujuan Cao, Qixiang Ye, Feiyue Huang, and David Doermann. Towards optimal structured cnn pruning via generative adversarial learning. In Proc. CVPR, pages 2790–2799, 2019.

[25] Hao Li, Asim Kadav, Igor Durdanovic, Hanan Samet, and Hans Peter Graf. Pruning filters for efficient convnets. In Proc. ICLR, 2017.

[51] Ruichi Yu, Ang Li, Chun-Fu Chen, Jui-Hsin Lai, Vlad I Morariu, Xintong Han, Mingfei Gao, Ching-Yung Lin, and Larry S Davis. NISP: Pruning networks using neuron importance score propagation. In Proc. CVPR, pages 9194–9203, 2018.

[35] Breton Minnehan and Andreas Savakis. Cascaded projection: End-to-end network compression and acceleration. In Proc. CVPR, June 2019.

[21] Hyeji Kim, Muhammad Umar Karim Khan, and Chong-Min Kyung. Efficient neural network compression. In Proc. CVPR, June 2019.

[14] Yihui He, Ji Lin, Zhijian Liu, Hanrui Wang, Li-Jia Li, and Song Han. AMC: AutoML for model compression and acceleration on mobile devices. In Proc. ECCV, pages 784–800, 2018.

[28] Yuchao Li, Shaohui Lin, Baochang Zhang, Jianzhuang Liu, David Doermann, Yongjian Wu, Feiyue Huang, and Rongrong Ji. Exploiting kernel sparsity and entropy for interpretable CNN compression. In Proc. CVPR, 2019.

[15] Yang He, Ping Liu, Ziwei Wang, Zhilan Hu, and Yi Yang. Filter pruning via geometric median for deep convolutional neural networks acceleration. In Proc. CVPR, pages 4340–4349, 2019. 3

---

### Author Response · Authors · 2021-08-25
**Additional comparison experiments on ResNet-164, CIFAR100**

We borrowed the below comparison table from Table 2 in [a].
In the experiment [a], an unpruned baseline(or a teacher) model was trained for 300 epochs, and fine-tuned for maximum 500 epochs. Thus, we simply trained our compressible model for 400 epochs. For our proposed method, we ran each experiment 5 times and the numbers are the average from these 5 repetitions.

.

Comparison of CIFAR100/ResNet-164 compression results. “FLOPs” and “Params” denote the remaining percentage of FLOP and parameter quantities of the compressed models and the lower the better.

.

Method$\space\space\space\space\space\space\space$ | Top-1/BL (%) | FLOPs (%) | Params (%)

SSS[19] $\space\space\space\space\space\space\space$|$\space\space$ 24.42/23.31 $\space\space$|$\space\space\space\space$ 55.33$\space\space$ |$\space\space\space$ 86.75

Hinge[a] $\space\space\space\space\space$|$\space\space$ 23.12/23.22 $\space\space$|$\space\space\space\space$ 55.32$\space\space$ |$\space\space\space$ 76.57

.

Our Proposed method.

No Distill. $\space\space\space\space$|$\space\space\space$ 23.98/22.34 $\space\space$|$\space\space\space\space$ 55.56$\space\space\space$ |$\space\space\space$ 75.03

With Distill.$\space$ |$\space\space\space$ 21.57/22.34 $\space\space$|$\space\space\space\space$ 54.97$\space\space\space$ |$\space\space\space$ 69.45

.

Our Baseline/Teacher Model (ResNet-164), trained for 300 epochs

Method$\space$ |$\space\space$Top-1 (%) $\space\space$ | $\space\space$ FLOPs $\space\space\space\space\space$  | Params

Baseline | 22.34 $\pm$ 0.26 | $4.97 \times 10^8$ | $1.73 \times 10^6$

.

Detailed Results for Proposed Method - without distillation, trained for 400 epochs

$\space\space\space\space\space\space\space$ $\lambda$ $\space\space\space\space\space\space\space\space\space$ | Top-1/BL (%)$\space$ | FLOPs (%) | Params (%)

$\lambda$=0.0000250$\space$|$\space\space\space$ 23.31/22.34 $\space\space$|$\space\space\space\space$ 65.05$\space\space\space$ |$\space\space\space$ 83.05

$\lambda$=0.0000500$\space$|$\space\space\space$ 23.98/22.34 $\space\space$|$\space\space\space\space$ 55.56$\space\space\space$ |$\space\space\space$ 75.03

$\lambda$=0.0000750$\space$|$\space\space\space$ 24.34/22.34 $\space\space$|$\space\space\space\space$ 45.29$\space\space\space$ |$\space\space\space$ 62.24

.

Detailed Results for Proposed Method - with distillation, $\alpha=0.4$ and $T=4$ in Eq. (9) in [a], trained for 400 epochs

$\space\space\space\space\space\space\space$ $\lambda$ $\space\space\space\space\space\space\space\space\space$ | Top-1/BL (%)$\space$ | FLOPs (%) | Params (%)



$\lambda$=0.0000500$\space$|$\space\space\space$ 21.23/22.34 $\space\space$|$\space\space\space\space$ 68.59$\space\space\space$ |$\space\space\space$ 85.48

$\lambda$=0.0000750$\space$|$\space\space\space$ 21.19/22.34 $\space\space$|$\space\space\space\space$ 61.35$\space\space\space$ |$\space\space\space$ 78.21

$\lambda$=0.0001000$\space$|$\space\space\space$ 21.57/22.34 $\space\space$|$\space\space\space\space$ 54.97$\space\space\space$ |$\space\space\space$ 69.45

$\lambda$=0.0001250$\space$|$\space\space\space$ 21.92/22.34 $\space\space$|$\space\space\space\space$ 48.01$\space\space\space$ |$\space\space\space$ 58.94

$\lambda$=0.0001500$\space$|$\space\space\space$ 22.33/22.34 $\space\space$|$\space\space\space\space$ 42.51$\space\space\space$ |$\space\space\space$ 48.67

.

[a]: Li, Yawei et al., Group Sparsity: The Hinge Between Filter Pruning and Decomposition for Network Compression, CVPR 2020

[19] Zehao Huang and Naiyan Wang. Data-driven sparse structure selection for deep neural networks. In Proc. ECCV, pages 304–320, 2018.

---

### Author Response · Authors · 2021-09-02
**Additional comparison experiments on ResNet-56, CIFAR100**

We borrowed the below comparison table from Table 1 in [e]. In the experiment [e], an unpruned baseline model was trained for 200 epochs, and fine-tuned for 200 epochs. We trained our baseline (teacher) model for 300 epochs and a student model (knowledge distillation) for 480 epochs. For our proposed method, we ran each experiment 5 times and the numbers are the average from these 5 repetitions.


Comparison of CIFAR100/ResNet-56 compression results. “FLOPs” and “Params” denote the remaining percentage of FLOP and parameter quantities of the compressed models and the lower the better.

.

Method$\space\space\space\space\space\space\space$ | Top-1/BL (%) | FLOPs (%) | Params (%)

NS[19] $\space\space\space\space\space\space\space$|$\space\space$ 28.60/27.51 $\space\space$|$\space\space\space\space$ 76.00$\space\space$ |$\space\space\space$ N/A

Polar[e] $\space\space\space\space\space$|$\space\space$ 27.54/27.51 $\space\space$|$\space\space\space\space$ 75.00$\space\space$ |$\space\space\space$ N/A

.

Our Proposed method.

With Distill.$\space$ |$\space\space\space$ 24.87/27.14 $\space\space$|$\space\space\space\space$ 64.59$\space\space\space$ |$\space\space\space$ 83.83


.

Our Baseline/Teacher Model (ResNet-56), trained for 300 epochs

Method$\space$ |$\space\space$Top-1 (%) $\space\space$ | $\space\space$ FLOPs $\space\space\space\space\space$  | Params

Baseline | 27.14 $\pm$ 0.30 | $2.52 \times 10^8$ | $8.59 \times 10^5$

.

Detailed Results for Proposed Method - with distillation, $\alpha=0.4$ and $T=4$ in Eq. (9) in [a], trained for 480 epochs

$\space\space\space\space\space\space\space$ $\lambda$ $\space\space\space\space\space\space\space\space\space$ | Top-1/BL (%)$\space$ | FLOPs (%) | Params (%)

$\lambda$=0.0001500$\space$|$\space\space\space$ 24.50/27.14 $\space\space$|$\space\space\space\space$ 73.77$\space\space\space$ |$\space\space\space$ 87.54

$\lambda$=0.0002000$\space$|$\space\space\space$ 24.87/27.14 $\space\space$|$\space\space\space\space$ 64.59$\space\space\space$ |$\space\space\space$ 83.83

$\lambda$=0.0002500$\space$|$\space\space\space$ 25.10/27.14 $\space\space$|$\space\space\space\space$ 65.18$\space\space\space$ |$\space\space\space$ 79.62

.

[a]: Li, Yawei et al., Group Sparsity: The Hinge Between Filter Pruning and Decomposition for Network Compression, CVPR 2020

[19] Zhuang Liu, Jianguo Li, Zhiqiang Shen, Gao Huang, Shoumeng Yan, and Changshui Zhang.
Learning efficient convolutional networks through network slimming. In ICCV, 2017.

[e] Zhuang et al., Neuron-level Structured Pruning using Polarization Regularizer, NeurIPS 20.

---

### Decision · Program_Chairs · 2021-09-27

**Decision:**

Reject

**Comment:**

This paper proposes a method based on sub-differential to sparsifying a network parameters during training. Although this paper contains novel interesting ideas, the results are not convincing nor significant enough to convince the reviewers. The reviews unanimously suggest the quality and significance of this paper is below the standard of the conference. The AC agrees with their recommendations.